# A Limit Load Solution for Anisotropic Welded Cracked Plates in Pure Bending

**Sergei Alexandrov [1,2,\*]**, **Elena Lyamina [1]**, **Alexander Pirumov [3]** and **Dinh Kien Nguyen [4,5]**

1   Ishlinsky Institute for Problems in Mechanics RAS, 119526 Moscow, Russia; lyamina@inbox.ru
2   Federal State Autonomous Educational Institution of Higher Education, South Ural State University (National Research University), 454080 Chelyabinsk, Russia
3   Department of Elektrotechnics and Mechanics, Russian Technological University (MIREA), 119454 Moscow, Russia; alpirumov@mail.ru
4   Institute of Mechanics, VAST, 18 Hoang Quoc Viet, Hanoi 100000, Vietnam; ndkien@imech.vast.vn
5   Faculty of Mechanics and Automation, Graduate University of Science and Technology, VAST, 18 Hoang Quoc Viet, Hanoi 100000, Vietnam
\*   Correspondence: sergei_alexandrov@spartak.ru

**Abstract:** The present paper's main objective is to derive a simple upper bound solution for a welded plate in pure bending. The plate contains a crack located in the weld. Both the weld and base materials are orthotropic. Hill's quadratic yield criterion is adopted. The solution is semi-analytic. A numerical method is only required for minimizing a function of two independent variables. Six independent dimensionless parameters classify the structure. Therefore, the complete parametric analysis of the solution is not feasible. However, for a given set of parameters, the numerical solution is straightforward, and the numerical method is fast. A numerical example emphasizes the effect of plastic anisotropy and the crack's location on the bending moment at plastic collapse. In particular, the bending moment for the specimen having a vertical axis of symmetry is compared with that of the asymmetric specimen. It is shown that the latter is smaller for all considered cases. The solution found can be used in conjunction with flaw assessment procedures.

**Keywords:** welded joints; cracks; limit load; anisotropy

## 1. Introduction

Welded joints are widely used in different sectors of the industry. Depending on the operating parameters, such joints' failure may initiate at the interface between the weld and base materials, the heat-affected zone, the base material, and other locations [1]. Moreover, various failure mechanisms are possible. The development of creep damage in the heat-affected zone has been studied in [2]. Corrosion fatigue experiments on butt welds of G20Mn5QT cast steel and Q345D hot rolled steel have been carried out in [3]. The creep crack path in P91 steel weldments has been investigated in [4]. A new cohesive interface model has been proposed in [5]. The model has been used for numerical simulation of weld fractures. The change between the interfacial failure mode and the pull-out failure mode of automotive steels' resistance spot welds has been studied in [6]. The hydrogen diffusion behavior during the welding of plates has been reported in [7]. It has been found that the position with the most severe cold cracking sensitivity is located at a 20–30 mm depth from the plate's surface. Plastic properties of bead-on-plate welds made using two types of seamless, copper-plated flux-cored wires have been determined in [8]. The present paper deals with the plastic collapse of welded joints.

Limit analysis is a crucial tool for studying the failure of structures for many applications [9–12]. The present paper focuses on the limit load for welded structures with cracks. The limit load is an

essential input parameter of commonly accepted flaw assessment procedures, determining the practical need for limit load solutions [13]. The theoretical basis for calculating limit loads is the lower and upper bound theorems. In the case of rigid perfectly plastic materials, these theorems can be found in any monograph on plasticity theory, for example, [14]. The upper bound theorem for more general material models has been proven in [15]. It is worthy of note that elastic properties do not affect the plastic limit load [16]. A recent review on limit load analysis has been provided in [17].

Most of the modern papers devoted to limit load solutions, including solutions for welded joints, are based on numerical methods [18–28]. Numerical methods are efficient for analyzing a given structure. However, limit analysis is also used to design structures, for example, [29]. A possible approach to design structures using numerical solutions can be based on fitting numerical results with analytic functions [20,30]. However, many structures are classified by so many parameters, meaning that this approach is not feasible. Therefore, analytic and semi-analytic solutions are preferable for a large class of structures [31]. It is worthy of note that the upper bound theorem's general mathematical features enable accurate predictions of the limit load even from comparatively crude approximations of the real velocity field [15]. Therefore, it is not surprising that numerical calculations usually confirm a high accuracy of analytic solutions, for example, [18,32]. A review of analytic limit load solutions for structures containing defects has been provided in [33].

It is known that plastic anisotropy has a significant effect on the predicted response of structures, for example, [34,35]. The effect of plastic anisotropy on the limit load has been emphasized in [36–39]. A numerical method for finding limit load solutions for structures made of plastically anisotropic materials has been developed in [40]. A review of limit load solutions for plastically anisotropic welded structures, including structures with cracks, has been provided in [41]. In particular, a solution for the pure bending of highly undermatched welded plates can be found in [42]. The present paper deals with an upper bound limit load solution for cracked welded plates in pure bending. It is assumed that both the weld and base materials are plastically anisotropic. The crack is located in the weld. The plate is or is not symmetric, depending on the location of the crack. The number of parameters classifying the structure significantly increases due to the weld containing the crack in conjunction with the anisotropic properties. Therefore, it is vital to find an analytic or semi-analytic solution to the problem formulated, as has already been noted above. The solution provided in the present paper is a generalization of the solution for isotropic materials given in [43].

## 2. Statement of the Problem

The geometry of the specimen is shown in Figure 1. The thickness of the specimen is $B$. It is assumed that the state of strain is plane. Therefore, $B$ is not involved in the solution. The thickness of the weld is $2W$, the specimen's width is $h$, and the length of the crack is $a$. The crack is parallel to the interfaces between the weld and base materials. It is located at the distance $t$ from the right interface, and $0 \le t \le W$. Thus, there are three essential dimensionless geometric parameters. The bending moment $M$ applies, as shown in Figure 1. It is necessary to find the value of $M$ at plastic collapse. This test and its theoretical description are used in conjunction with flow assessment procedures [13,30].

It is assumed that the weld and base materials are orthotropic. The straight principal axes of anisotropy are parallel and perpendicular to the weld and base material interfaces. Each material obeys Hill's quadratic yield criterion [14]. Let $(x, y)$ be a Cartesian coordinate system whose coordinate lines coincide with the principal axes of anisotropy. Then, the plane strain yield criterion reads

$$\frac{\left(\sigma_{xx} - \sigma_{yy}\right)^2}{4(1-c)} + \sigma_{xy}^2 = T^2. \tag{1}$$

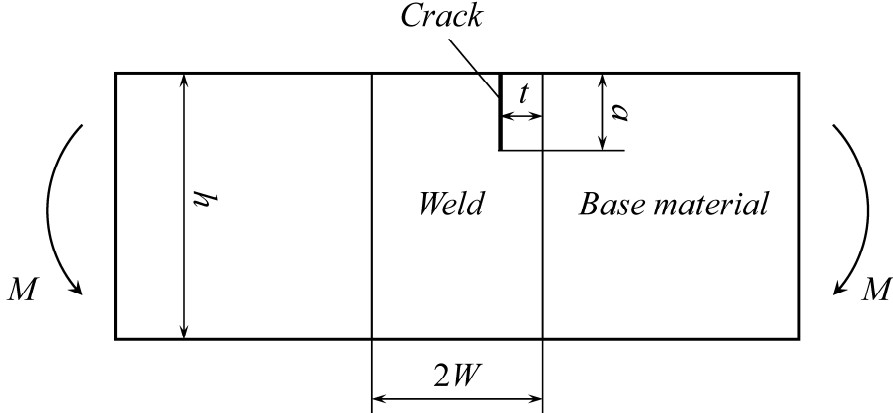

**Figure 1.** Geometry of the specimen.

Here, $\sigma_{xx}$, $\sigma_{yy}$, and $\sigma_{xy}$ are the components of the stress tensor referred to the $(x, y)-$ coordinate system, $T$ is the shear yield stress in this coordinate system, and $c$ is a parameter that can be expressed in terms of the tensile yield stresses in the directions of the principal axes of anisotropy and $T$. One can find the precise expressions in [14]. For the purpose of the present study, it is sufficient to know that $1 > c > -\infty$. It is evident from (1) that $c$ is dimensionless. In what follows, the subscript "$w$" will denote the weld's mechanical properties, $c_w$ and $T_w$, and the subscript "$b$" will denote the mechanical properties of the base material, $c_b$ and $T_b$. Thus, there are three essential dimensionless physical parameters. In total, six dimensionless parameters affect the bending moment at plastic collapse.

## 3. General Solution

### 3.1. Kinematically Admissible Velocity Field

It is assumed that the specimen consists of two rigid blocks. One of these blocks is motionless. The other rotates with an angular velocity $w$ around point $O$. The interface between the blocks is a velocity discontinuity line. This line is a circular arc with its center at $O$. The location of point $O$ and the radius of the velocity discontinuity line, $\rho$, should be found from the solution. The velocity discontinuity line must pass through the crack tip and reach the lower surface of the specimen.

There are three possible flow patterns. These patterns result in different expressions for the upper bound bending moment. The first flow pattern is shown in Figure 2. Its distinguishing feature is that the velocity discontinuity line intersects the right interface between the weld and base materials (Figure 2a) or the left interface (Figure 2b) at two points. These two cases are identical if the specimen has a vertical axis of symmetry (i.e., $t = W$ in Figure 1). The location of point $O$ is determined by $X_0$ and $Y_0$. The other geometric parameters that affect the upper bound bending moment are $\rho$ and the angles $\theta_1$, $\theta_2$, $\theta_3$, and $\theta_4$.

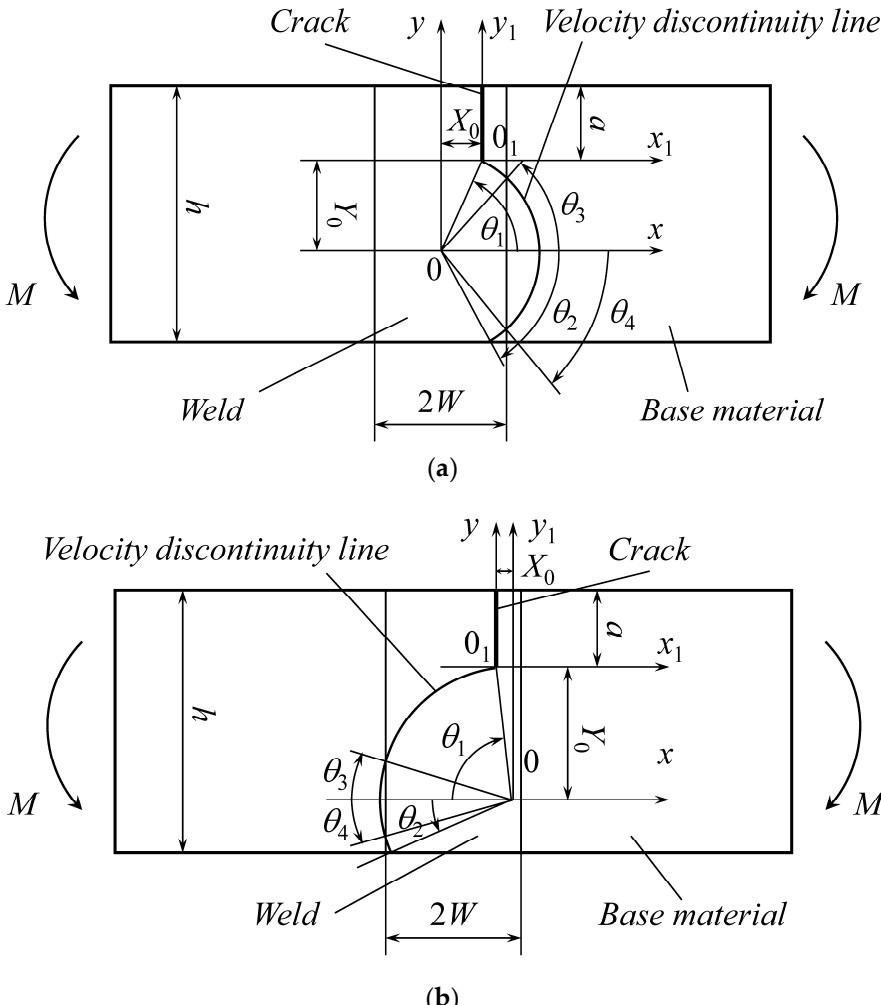

**Figure 2.** First pattern of the kinematically admissible velocity field on the right (**a**) and left (**b**) to the crack line.

The second flow pattern is shown in Figure 3. Its distinguishing feature is that the velocity discontinuity line does not intersect the interfaces between the weld and base materials. The cases illustrated in Figure 3a,b are identical if $t = W$. Otherwise, it is necessary to consider both cases. Five geometric parameters are involved in the description of this flow pattern, which are $\rho$, $X_0$, $Y_0$, $\theta_1$, and $\theta_2$.

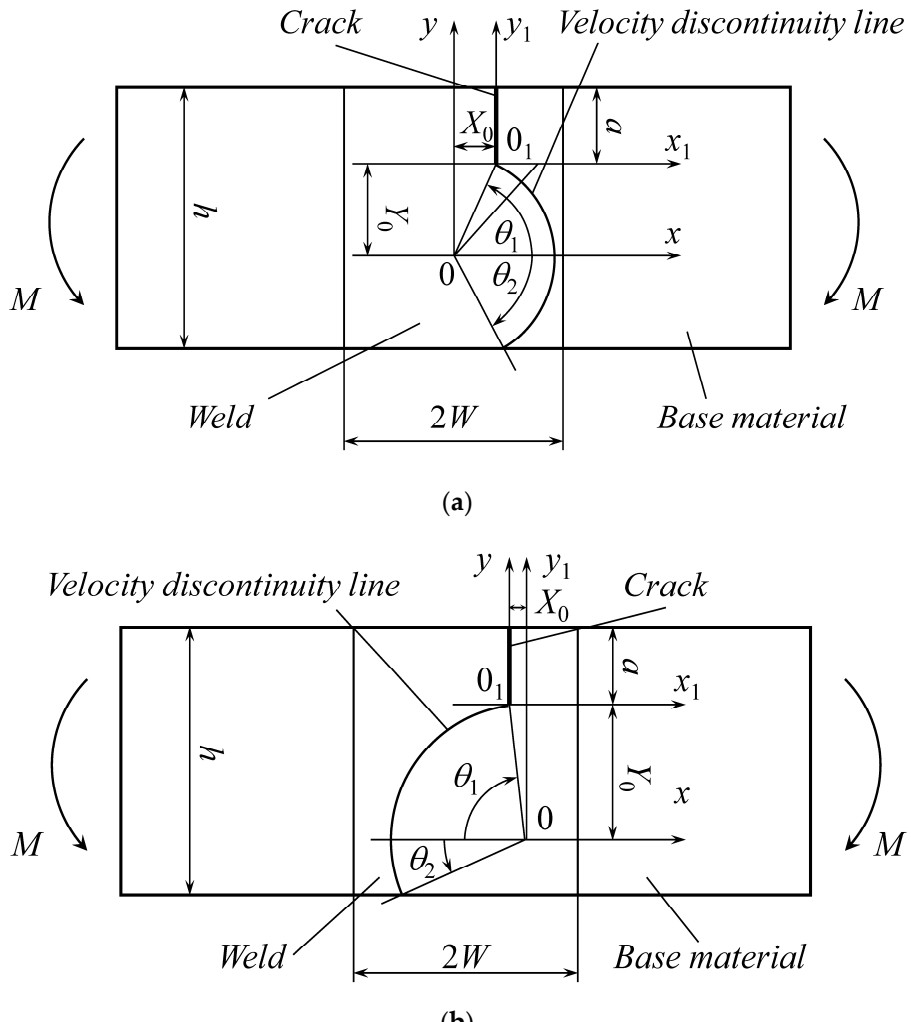

**Figure 3.** Second pattern of the kinematically admissible velocity field on the right (**a**) and left (**b**) to the crack line.

The third flow pattern is shown in Figure 4. Its distinguishing feature is that the velocity discontinuity line intersects the right interface between the weld and base materials (Figure 4a) or the left interface (Figure 4b) at one point. The cases illustrated in Figure 4a,b are identical if $t$ = $W$. Otherwise, it is necessary to consider both cases. Six geometric parameters are involved in the description of this flow pattern, which are $\rho$, $X_0$, $Y_0$, $\theta_1$, $\theta_2$, and $\theta_3$.

The upper bound theorem allows for determining which of the patterns above and which case for each pattern result in a better approximation of the plastic collapse's bending moment.

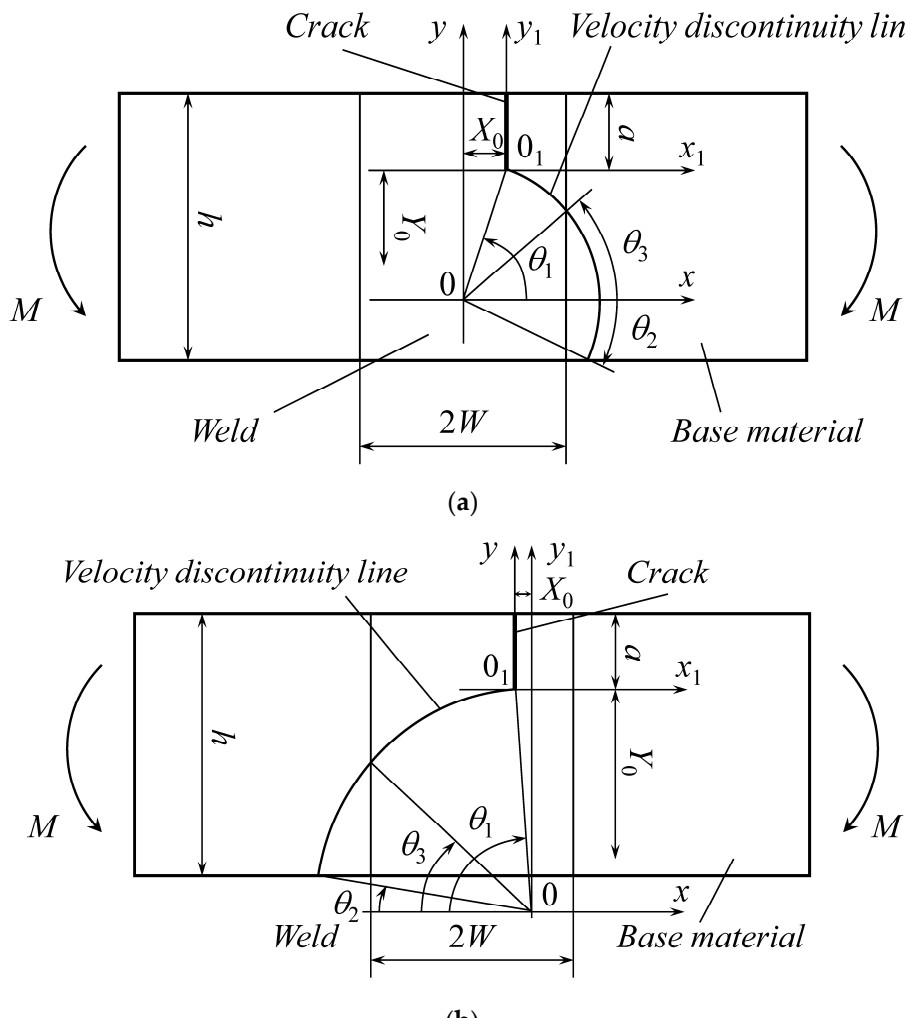

**Figure 4.** Third pattern of the kinematically admissible velocity field on the right (**a**) and left (**b**) to the crack line.

### 3.2. Relations between Geometric Parameters

Each of the flow patterns above involves several geometric parameters. However, only two of these parameters are independent. It is convenient to introduce an additional geometric parameter $b$, which is the distance between the line of the crack and the intersection point of the velocity discontinuity line and the lower surface of the specimen (Figure 5). One can take $b$ and $\rho$ as the independent geometric parameters. The relations between the independent and dependent geometric parameters are derived from Figure 5, which is self-explanatory, using simple geometric and trigonometric formulae. In particular,

$$X_0 = \rho \sin(\gamma + \alpha) - b, \quad Y_0 = \rho \cos(\alpha - \gamma),$$

$$\theta_1 = \begin{cases} X_0 < 0, \ \pi - \arcsin(Y_0/\rho) \\ X_0 \geq 0, \ \arcsin(Y_0/\rho) \end{cases}, \quad \theta_2 = \arcsin\left(\frac{a + Y_0 - h}{\rho}\right), \quad \theta_3 = \arccos\left(\frac{X_0 + t}{\rho}\right), \quad \theta_4 = -\theta_3. \tag{2}$$

Here,

$$\alpha = \arcsin\left(\frac{\sqrt{(h-a)^2 + b^2}}{2\rho}\right), \quad \gamma = \arcsin\left(\frac{h-a}{\sqrt{(h-a)^2 + b^2}}\right). \tag{3}$$

It is also seen from Figure 5 that $b \geq 0$ and $2\rho \geq \sqrt{(h-a)^2 + b^2}$. These inequalities are useful when the upper bound theorem applies.

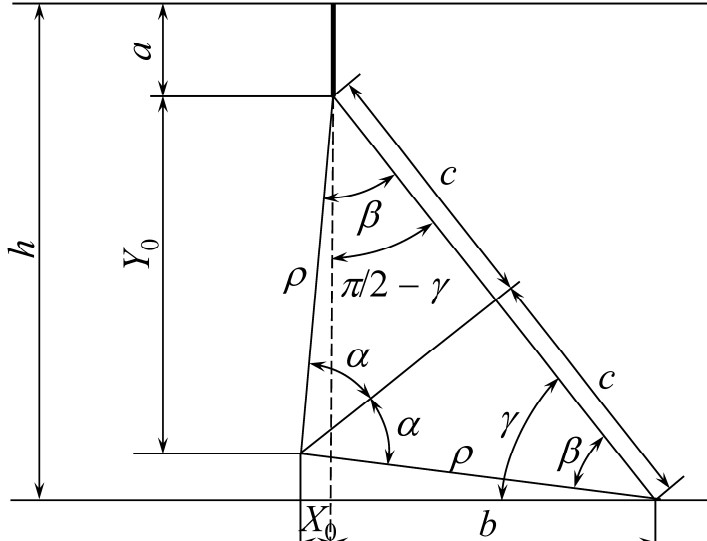

**Figure 5.** Illustration of the geometric parameters involved in the description of the kinematically admissible velocity field.

### 3.3. Bending Moment

It follows from the upper bound theorem [14] and the kinematically admissible velocity field chosen that

$$\mathrm{M}\omega \leq \mathrm{B}\rho \int_{\theta_2}^{\theta_1} [u] k \mathrm{d}\theta. \tag{4}$$

Here, $[u]$ is the amount of velocity jump across the velocity discontinuity line, $k$ is the shear stress acting on the velocity discontinuity line, and $\theta$ is the angle measured anticlockwise from the horizontal line through point $O$ (Figures 2–4). Since one of the rigid blocks is motionless and the other rotates with the angular velocity $\omega$,

$$[u] = \rho\omega. \tag{5}$$

The value of $k$ is determined as [14]

$$k = T_w \sqrt{1 - c_w \sin^2 2\theta} \tag{6}$$

for the portion of the velocity discontinuity line within the weld, and as

$$k = T_b \sqrt{1 - c_b \sin^2 2\theta} \tag{7}$$

for the portion of the velocity discontinuity line within the base material. Substituting Equations (5) to (7) into Equation (4) gives

$$m_u = \frac{M_u}{Bh^2 T_w} = \frac{\rho^2}{2h^2}[E(2\theta_1, c_w) - E(2\theta_3, c_w) + E(2\theta_4, c_w) - E(2\theta_2, c_w)] + $$
$$\frac{\rho^2}{2h^2}\frac{T_b}{T_w}[E(2\theta_3, c_b) - E(2\theta_4, c_b)] \tag{8}$$

for the first flow pattern (Figure 2),

$$m_u = \frac{M_u}{Bh^2 T_w} = \frac{\rho^2}{2h^2}[E(2\theta_1, c_w) - E(2\theta_2, c_w)] \tag{9}$$

for the second flow pattern (Figure 3), and

$$m_u = \frac{M_u}{Bh^2 T_w} = \frac{\rho^2}{2h^2}[E(2\theta_1, c_w) - E(2\theta_3, c_w)] + \frac{\rho^2}{2h^2}\frac{T_b}{T_w}[E(2\theta_3, c_b) - E(2\theta_2, c_b)] \tag{10}$$

for the third flow pattern (Figure 4). Here, $M_u$ is the upper bound on the bending moment, $m_u$ is its dimensionless representation, and $E(x, m)$ is the elliptic integral of the second kind. The latter is defined as

$$E(z, m) = \int \sqrt{1 - m \sin^2 z}\, dz. \tag{11}$$

It is seen from (2) and (3) that the right-hand sides of Equations (8) to (10) depend on $b$ and $\rho$. For finding the best upper bound, one should minimize $m_u$ with respect to these parameters. This part of the solution should be carried out numerically.

## 4. Numerical Example

The bending moment depends on six dimensionless parameters. Therefore, the complete parametric analysis of the solution derived in the previous section is not feasible. The numerical results below emphasize the effect of plastic anisotropy and the crack's location on the bending moment at plastic collapse. These results have been derived by minimizing the value of $m_u$ given in Equations (8)–(10). In all calculations, $W/h = 0.05$, $T_w/T_b = 2$, and $c_w = -6$.

Figure 6 depicts the dependence of $m_u$ on $c_b$ for several values of $a$ at $t = W$ (i.e., the specimen has a vertical axis of symmetry coinciding with the crack line). Figure 7 shows the dependence of $m_u$ on $c_b$ for several values of $a$ at $t = 0$ (i.e., the crack is located at the interface between the weld and base materials).

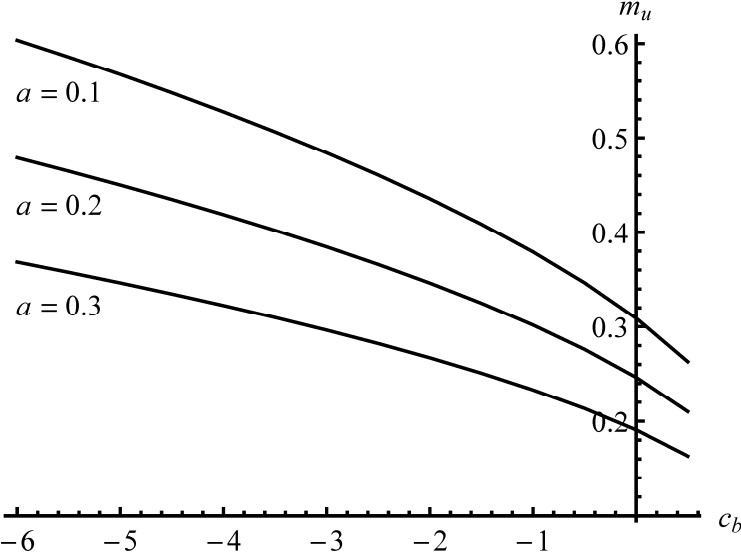

**Figure 6.** Variation of $m_u$ with $c_b$ for several values of $a$ at $c_w = -6$, $W/h = 0.05$, $T_w/T_b = 2$, and $t = W$ (symmetric case).

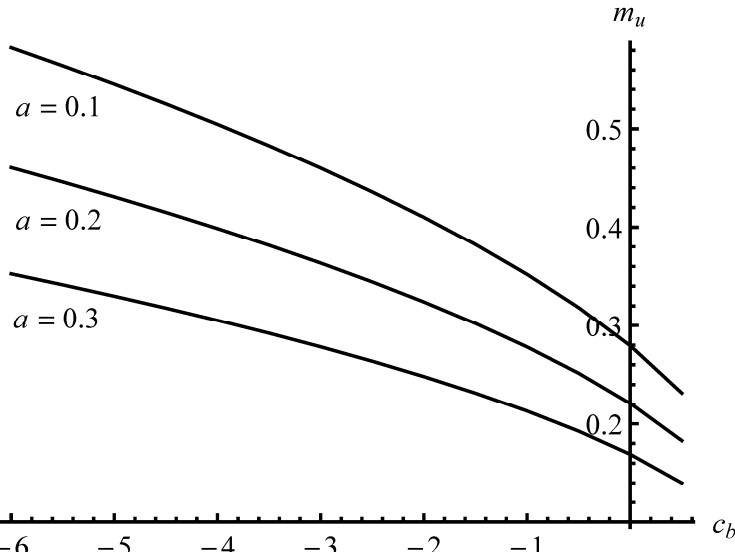

**Figure 7.** Variation of $m_u$ with $c_b$ for several values of $a$ at $c_w = -6$, $W/h = 0.05$, $T_w/T_b = 2$, and $t = 0$.

It is seen from Figures 6 and 7 that the bending moment increases as $c_b$ decreases, and decreases as $a$ increases. In particular, these results show a significant effect of plastic anisotropy on the bending moment at plastic collapse. They also show that the crack located at the interface (asymmetric case) is less safe than the same crack at the center (symmetric case).

## 5. Summary

An upper bound limit load solution for the specimen described in Section 2 has been derived. Three different flow patterns of the kinematically admissible velocity field are possible. These patterns are illustrated in Figures 2–4. The velocity discontinuity line may or may not intersect the interface between the weld and base materials. There are two intersection points in the case of the first flow pattern, one point in the third flow pattern, and no point in the second flow pattern. Accordingly, there are three different formulae for the bending moment at plastic collapse. These formulae are given in Equations (8) to (10). The solution found is a generalization of the solution provided in [43] on orthotropic materials. This available solution is obtained from the new solution if $c_w = c_b = 0$ and $t = W$.

Equations (8) to (10) involve two independent parameters, $b$ and $\rho$. These parameters should be found by minimizing the right-hand sides of these equations. At this stage of the solution, the operative flow pattern for a given plate and the best upper bound on the bending moment at plastic collapse are determined. This result is final for plates that have a vertical axis of symmetry coinciding with the crack line. For asymmetric plates, the velocity discontinuity line may be located on the right or the left of the crack line (Figure 2a,b, Figure 3a,b, and Figure 4a,b). Therefore, one should repeat the procedure above two times. As a result, two values for the upper bound of the bending moment at plastic collapse are found for the same plate. The smaller should be chosen as the final result.

Equations (8) to (10) are combinations of analytic functions. However, six independent parameters classify the plate (three geometric parameters and three physical parameters). Therefore, the complete parametric analysis of the solution is not feasible. However, for any given set of parameters, the solution is straightforward, and the numerical minimization is fast. Therefore, the solution is suitable for a quick assessment of cracked plates using flaw assessment procedures [13].

The numerical example in Section 4 emphasizes the effect of plastic anisotropy and the crack's location on the bending moment at plastic collapse (Figures 6 and 7). It is seen from these figures that the bending moment increases as $c_b$ decreases. That is, materials with a smaller $c_b$ increase the resistance of the plate. It is also seen from Figures 6 and 7 that a crack at the interface between

the weld and base materials (asymmetric plates) is less safe than the same crack at the center (symmetric plates). It is worthy to note that these conclusions are valid only for the range of parameters used in the numerical solution.

**Author Contributions:** Conceptualization, S.A.; writing, A.P.; formal analysis, E.L., numerical method, D.K.N. All authors have read and agreed to the published version of the manuscript.

**Funding:** This research was made possible by the grants RFRR-20-51-54005 (Russia). QTRU01.07/20-21 (Vietnam), and AAAA-A20-120011690136-2 (Russia).

**Conflicts of Interest:** The authors declare no conflict of interest.

## Nomenclature

| | |
|---|---|
| $a$ | crack length |
| $c$ | constitutive parameter introduced in Equation (1) |
| $c_b$ | value of $c$ for the base material |
| $c_w$ | value of $c$ for the weld material |
| $h$ | specimen width |
| $m_u$ | dimensionless upper bound on the bending moment |
| $(x, y)$ | Cartesian coordinates |
| $B$ | specimen thickness |
| $M$ | bending moment |
| $M_u$ | upper bound on the bending moment |
| $T$ | shear yield stress in the Cartesian coordinates |
| $T_b$ | value of T for the base material |
| $T_w$ | value of T for the weld material |
| $W$ | half-thickness of weld |
| $X_0, Y_0$ | parameters determining the position of point 0 relative to the crack tip (Figure 2). |
| $\theta_1, \theta_2, \theta_3, \theta_4$ | angles introduced in Figure 2 |
| $\rho$ | radius of velocity discontinuity lines |
| $\sigma_{xx}, \sigma_{yy}, \sigma_{xy}$ | stress components in the Cartesian coordinates |

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
