# Peer review of "A Limit Load Solution for Anisotropic Welded Cracked Plates in Pure Bending"

_symmetry, doi:10.3390/sym12111764_

Round 1

Reviewer 1 Report

Dear Authors,

I have read Your submission with great attention. I believe that the analysis of the effect of cracks in welded joints subjected to operational loads should be considered, as it enables prediction of the service life of welded structures. In my opinion, the work has the right structure, is well thought out in all respects and the conclusions I consider important. I am convinced that the research material described in the manuscript will be an important source of information for scientists and (to a certain extent ) for industrial engineers. While reading the article, I had a few comments that, hopefully, will help shape the final version of your article.

  1. There is no information on the material of the analysis. I propose to mention in the abstract and chapter 2 what material limitations the solution under consideration has. Could you specify this on line 189 with another limitation?
  2. The Introduction section is short, it actually focuses only on showing the state of knowledge regarding research methodology. I propose to extend this chapter with a paragraph describing the role of imperfections in welded joints: in the weld, HAZ and base material, which can be divided into technological (cold cracking, hot cracking, lamellar tearing and reheat cracking) and operational (hydrogen embrittlement, fatigue cracking, corrosion cracking etc). In this regard, I can recommend current articles from the MDPI publishing house: https://doi.org/10.3390/ma13173888; https://doi.org/10.3390/ma13173887. I am convinced that mentioning the practical application of the research method and citing articles from MDPI will have a positive impact on readership and citation of the article.
  3. Lines 55-61: please note that this fragment of the work is close to Chapter 2. I think it should be moved further.
  4. Line 55: fragment: “solution for welded plates with a crack in pure bending” may be misunderstood: “crack in pure bending”. Please write this sentence in a different way.
  5. I also have a debatable question, from a practical point of view: which welded joints are subject to the load shown in Figure 1 (maybe ship's side vertical joints?)? Chapter 2 presents the assumptions, maybe it is worth expanding them with this aspect?
  6. Figures 2, 3 and 4 are square so I suggest you turn them 90 degrees. This will allow for a similar orientation to Figure 1, which will facilitate their comparison.
  7. Chapter 4: What tool (software) was used to carry out these analyzes?
  8. According to the MDPI Guidelines, “Conclusions” as proposed should be entitled "Summary". This section does not compare the results with the state of knowledge in the literature. I believe that this should be completed in order to get a complete picture of the issue.
  9. References are remarkably well matched (up to date and from serious, credible sources), but lacking MDPI articles.

Author Response

All corrections are shown in red.

1. There is no information on the material of the analysis. I propose to mention in the abstract and chapter 2 what material limitations the solution under consideration has. Could you specify this on line 189 with another limitation?

Response. Do you mean the physical material? We work with a model, not with real material. The solution is valid for any material that satisfies (1). We are afraid that we do not understand this comment. Please clarify.

2. The Introduction section is short, it actually focuses only on showing the state of knowledge regarding research methodology. I propose to extend this chapter with a paragraph describing the role of imperfections in welded joints: in the weld, HAZ and base material, which can be divided into technological (cold cracking, hot cracking, lamellar tearing and reheat cracking) and operational (hydrogen embrittlement, fatigue cracking, corrosion cracking etc). In this regard, I can recommend current articles from the MDPI publishing house: https://doi.org/10.3390/ma13173888; https://doi.org/10.3390/ma13173887. I am convinced that mentioning the practical application of the research method and citing articles from MDPI will have a positive impact on readership and citation of the article.

Response. We have added several references, including those recommended.

3. Lines 55-61: please note that this fragment of the work is close to Chapter 2. I think it should be moved further.

Response. Yes, we agree that this portion of the Introduction section is a short description of the problem solved. However, each portion of the Introduction section related to the research carried out in the paper is a short description of the other sections' contents.

4. Line 55: fragment: “solution for welded plates with a crack in pure bending” may be misunderstood: “crack in pure bending”. Please write this sentence in a different way.

Response. We have corrected it.

5. I also have a debatable question, from a practical point of view: which welded joints are subject to the load shown in Figure 1 (maybe ship's side vertical joints?)? Chapter 2 presents the assumptions, maybe it is worth expanding them with this aspect?

Response. This test is used in conjunction with flaw assessment procedures. We have added this point to Section 2.

6. Figures 2, 3 and 4 are square so I suggest you turn them 90 degrees. This will allow for a similar orientation to Figure 1, which will facilitate their comparison.

Response. We have corrected figures 2 to 4.

7. Chapter 4: What tool (software) was used to carry out these analyzes?

Response. MATHEMATICA (https://www.wolfram.com/). It is not important which software to use. One may use MAPLE (https://www.maplesoft.com/), for example.

8. According to the MDPI Guidelines, “Conclusions” as proposed should be entitled "Summary". This section does not compare the results with the state of knowledge in the literature. I believe that this should be completed in order to get a complete picture of the issue.

Response. We have corrected this section. In particular, comparison with the solution in [43] is included in this section.

9. References are remarkably well matched (up to date and from serious, credible sources), but lacking MDPI articles.

Response. In the revised manuscript, the references to MDPI publications are [7], [8], [28], [41].

Reviewer 2 Report

The authors present the semi-analytical solution of the welded plate in pure bending. The plate contains a crack located in the weld. The effect of plastic anisotropy and the crack's location on the bending moment at plastic collapse was considered in case of the numerical example. Tne paper present only numerical solution of the welded plate in pure bending without the experiment.
I propose to prove the accuracy of the presented solutions by experiment.

Author Response

The authors present the semi-analytical solution of the welded plate in pure bending. The plate contains a crack located in the weld. The effect of plastic anisotropy and the crack's location on the bending moment at plastic collapse was considered in case of the numerical example. Tne paper present only numerical solution of the welded plate in pure bending without the experiment.
I propose to prove the accuracy of the presented solutions by experiment.

Response.

Our area of expertise is the theory. We are not capable of doing any experiment.

Yes, of course, it would be nice if every paper published in the literature includes both the theory and experiment. Unfortunately, it is not realistic. In most cases, papers that develop new theoretical methods do not include experiments, and papers that develop new experimental techniques do not include the theory (except trivial FE solutions). In particular, most papers in the list of references to our paper are purely theoretical.

We submitted the paper to the special issue https://www.mdpi.com/journal/symmetry/special_issues/Plastically_Anisotropic_Materials_2020.

The special issue information reads

This Special Issue of Symmetry features articles about analytical and numerical methods of analysis and design of structures and machine parts, assuming that the material is plastically anisotropic.

As you can see, experimental research is not included.

Reviewer 3 Report

Article by title  "A Limit Load Solution for Anisotropic Welded Cracked Plates in Pure Bending" is well presented.

However, there are some minor bugs that need to be corrected.
After corrections are made, the article is ready for publication.

Figure 7. should be moved to Chapter 3.

It would be useful to add a list of the most important designations at the end of the manuscript. This will make the article more readable.

Author Response

All corrections are shown in red.

  1. Figure 7. should be moved to Chapter 3

We are afraid that we do not understand this comment. Figure 7 appears for the first time in Section 4. Why do you think that the figure should be moved to Section 3?.

  1. It would be useful to add a list of the most important designations at the end of the manuscript. This will make the article more readable.

We have added the list of symbols.

Round 2

Reviewer 1 Report

Dear Authors,

thank you for the answers to my doubts and the appropriate explanations that will make the work less hermetic. I believe that publishing the article will be beneficial for the scientific community.

Finally, I would like to mention that it is clear to me that the profile of the journal justifies the format of your article - no experiment. Best regards